# Learning to Decouple Complex System for Sequential Data

## Abstract

A complex system with cluttered observations may be a coupled mixture of multiple simple sub-systems corresponding to *latent entities*. Such sub-systems may hold distinct dynamics in the continuous-time domain, therein complicated interactions between sub-systems also evolve over time. This setting is fairly common in the real world, but has been less considered. In this paper, we propose a sequential learning approach under this setting by decoupling a complex system for handling irregularly sampled and cluttered sequential observations. Such decoupling brings about not only subsystems describing the dynamics of each latent entity, but also a meta-system capturing the interaction between entities over time. Specifically, we argue that the meta-system of interactions is governed by a smoothed version of *projected differential equations*. Experimental results on synthetic and real-world datasets show the advantages of our approach when facing complex and cluttered sequential data compared to the state-of-the-art.

## 1 Introduction

Discovering hidden rules from sequential observations has been an essential topic in machine learning, with a large variety of applications such as physics simulation (Sanchez-Gonzalez et al., 2020), autonomous driving (Diehl et al., 2019), ECG analysis (Golany et al., 2021) and event analysis (Chen et al., 2021), to name a few. A standard scheme is to consider sequential data at each timestamp being holistic and homogeneous under some ideal assumptions (i.e., only the temporal behavior of one entity is involved in a sequence), under which data/observation is treated as collection of slices at different time from a unified system. A series of sequential learning models fall into this category, including variants of recurrent neural networks (RNNs) (Cho et al., 2014; Hochreiter & Schmidhuber, 1997), neural differential equations (DEs) (Chen et al., 2018; Kidger et al., 2020; Rusch & Mishra, 2021; Zhu et al., 2021) and spatial/temporal attention-based approaches (Vaswani et al., 2017; Fan et al., 2019; Song et al., 2017). These variants fit well into the scenarios agreeing with the aforementioned assumptions, and are proved effective in learning or modeling for relatively simple applications with clean data source.

In the real world, a system may not only describe a single and holistic entity, but also consist of several *distinguishable* interacting but simple subsystems, where each subsystem corresponds to a physical entity. For example, we can think of the movement of a solar system being the mixture of distinguishable subsystems of sun and surrounding planets, while interactions between these celestial bodies along time are governed by the laws of gravity. Back to centuries ago, physicists and astronomers made enormous effort to discover the rule of celestial movements from the records of each single bodies, and eventually delivered the neat yet elegant differential equations (DEs) depicting principles of moving bodies and interactions therein. Likewise in nowadays, researchers also developed a series of machine learning models for sequential data with distinguishable partitions (Qin et al., 2017). Two widely adopted strategies for learning the interactions between subsystems are graph neural networks (Iakovlev et al., 2021; Ha & Jeong, 2021; Kipf et al., 2018; Yıldız et al., 2022; Xhonneux et al., 2020) and attention mechanism (Vaswani et al., 2017; Lu et al., 2020; Goyal et al., 2021), while the interactions are typically encoded with "messages" between nodes and pair-wise "attention scores", respectively.

It is worth noting a even more difficult scenario, in which *the data/observation is so cluttered that cannot be readily distinguished into separate parts*. This can be either due to the way of data

collection (e.g., video consisting of multiple objects), or because there is no explicit physical entities originally (e.g., weather time series). To tackle this, a fair assumption can be introduced that complex observations can be decoupled into several relatively independent modules in the feature space, where each module corresponds to a *latent entity*. Latent entities may not have exact physical meanings, but learning procedures can greatly benefit from such decoupling, as this assumption can be viewed as strong regularization to the system. This assumption has been successfully incorporated in several models for learning from *regularly* sampled sequential data, by emphasizing "independence" to some extent between channels or groups in the feature space (Li et al., 2018; Yu et al., 2020; Goyal et al., 2021; Madan et al., 2021). Another successful counterpart in parallel benefiting from this assumption is transformer (Vaswani et al., 2017) which stacks multiple layers of self-attention and point-wise feedforward networks. In transformers, each attention head can be viewed as a relatively independent module and interaction happens throughout head re-weighting procedure following the attention scores. Lu et al. (2020) presented an interpretation from a dynamic point of view, by regarding a basic layer in the transformer as one step of integration, governed by differential equations derived from interacting particles. Vuckovic et al. (2020) extended this interpretation with more solid mathematical support by viewing forward pass of transformer as applying successive Markov kernels in a particle-based dynamic system.

We note, however, despite the ubiquity of this setting, there is barely any previous investigation focusing on learning for *irregularly sampled* and *cluttered* sequential data. Aforementioned works either fail to handle the irregularity (Goyal et al., 2021; Li et al., 2018), or neglect the independence/modularity assumption in the latent space (Chen et al., 2018; Kidger et al., 2020). In this paper, inspired by recent advances of neural controlled dynamics (Kidger et al., 2020) and novel interpretation of attention mechanism (Vuckovic et al., 2020), we make a step to propose an effective approach addressing this problem under dynamic setting. To this end, our approach explicitly learned to decouple a complex system into several latent sub-systems and utilizes an additional meta-system capturing the evolution of interactions over time. Specifically, taking into account the constrained interactions analogous to the attention mechanism, we further characterized such interactions using projected differential equations (ProjDEs). We argued our **contributions** as follows:

- We provide a novel modeling strategy for sequential data from a system decoupling perspective;
- We propose a novel and natural interpretation of evolving interactions as a ProjDE-based meta-system under constraints;
- Our approach is parameter-insensitive and more compatible to other modules, thus being flexible to be integrated into various tasks.

Extensive experiments were conducted on either regularly or irregularly sampled sequential data, including both synthetic and real-world settings. It was observed that our approach achieved prominent performance compared to state-of-the-arts on a wide spectrum of tasks.

## 2 RELATED WORK

**Sequential learning.** Traditionally, learning with sequential data can be performed using variants of recurrent neural networks (RNNs) (Hochreiter & Schmidhuber, 1997; Cho et al., 2014; Li et al., 2018) under a Markov setting. While such RNNs are generally designed for regularly sampling frequency, a more natural line of counterparts lie in the continuous time domain allowing irregularly sampled time series as input. As such, a variety of RNN-based methods are developed, by introducing exponential decay on observations (Che et al., 2018; Mei & Eisner, 2017), incorporating an underlying Gaussian process (Li & Marlin, 2016; Futoma et al., 2017), or integrating some latent evolution under ODEs (Rubanova et al., 2019; De Brouwer et al., 2019). A seminal work interpreting forward passing in neural networks as integration of ODEs was proposed in Chen et al. (2018), following by a series of relevant works (Liu et al., 2019; Li et al., 2020a; Dupont et al., 2019). As integration over ODEs allows for arbitrary step length, it's a natural modeling of irregularly time series, and proved powerful in many machine learning tasks (e.g., bioinformatics (Golany et al., 2021), physics (Nardini et al., 2021) and computer vision (Park et al., 2021)). Kidger et al. (2020) studied a more effective way of injecting observations to the system via a mathematical tool called Controlled differential Equation, achieving state-of-the-art performance on several benchmarks. Some variants of neural ODEs have

also been extended to discrete structure (Chamberlain et al., 2021a; Xhonneux et al., 2020) and non-Euclidean setting (Chamberlain et al., 2021b).

**Learning with independence.** Independence or modular property serve as strong regularization or prior in some learning tasks under static setting (Wang et al., 2020; Liu et al., 2020). In the sequential case, some early attempts over RNNs emphasized implicit "independence" in the feature space between dimensions or channels (Li et al., 2018; Yu et al., 2020). As independence assumption commonly holds in vision task (with distinguishable objects), Pang et al. (2020); Li et al. (2020b) proposed video understanding schemes by decoupling the spatio-temporal patterns. For a more generic case where the observations are collected without any prior, Goyal et al. (2021) devised an sequential learning scheme called recurrent independence mechanism (RIM), and its generalization ability was extensively studied in Madan et al. (2021). Lu et al. (2020) investigated self-attention mechanism (Vaswani et al., 2017) and interpreted it as a nearly independent multi-particle system with interactions therein. Vuckovic et al. (2020) further provided more solid mathematical analysis with the tool of Markov kernel. Study of such mechanism in the dynamical setting was barely observed.

## 3 METHODOLOGY

### 3.1 BACKGROUND

In this section, we briefly review three aspects related to our approach. Our approach is built upon the basic sub-system derived from *Neural Controlled Dynamics* (Kidger et al., 2020), while the interactions are modeled at an additional meta-system analogous to *Self-attention* (Lu et al., 2020; Vuckovic et al., 2020), and further interpreted and generalized using the tool of *Projected Differential Equations* (Dupuis & Nagurney, 1993).

**Neural Controlled Dynamics.** Continuous-time dynamics can be expressed using differential equations $\mathbf{z}'(t) = d\mathbf{z}/dt = f(\mathbf{z}(t), t)$, where $\mathbf{z} \in \mathbb{R}^d$ and $t$ are a $d$-dimension state and the time, respectively. Function $f : \mathbb{R}^d \times \mathbb{R}_+ \to \mathbb{R}^d$ governs the evolution of the dynamics. Given the initial state $\mathbf{z}(t_0)$, the state at any time $t_1$ can be evaluated with:

$$\mathbf{z}(t_1) = \mathbf{z}(t_0) + \int_{t_0}^{t_1} f(\mathbf{z}(s), s)\mathrm{d}s \tag{1}$$

In practice, we aim at learning the dynamics from a series of observations or controls $\{\mathbf{x}(t_k) \in \mathbb{R}^b | k = 0, 1, ...\}$ by parameterizing the dynamics with $f_\theta(\cdot)$ where $\theta$ is the unknown parameter to be learned. Thus, a generic dynamics incorporating outer signals $\mathbf{x}$ can be written as:

$$\mathbf{z}(t_1) = \mathbf{z}(t_0) + \int_{t_0}^{t_1} f_\theta(\mathbf{z}(s), \mathbf{x}(s), s)\mathrm{d}s \tag{2}$$

Rather than directly injecting $\mathbf{x}$ as in Eq. (2), Neural Controlled Differential Equation (CDE) proposed to deal with outer signals with a Riemann–Stieltjes integral (Kidger et al., 2020):

$$\mathbf{z}(t_1) = \mathbf{z}(t_0) + \int_{t_0}^{t_1} \mathbf{F}_\theta(\mathbf{z}(s))\mathbf{x}'(s)\mathrm{d}s \tag{3}$$

where $\mathbf{F}_\theta : \mathbb{R}^d \to \mathbb{R}^{d \times b}$ is a learnable vector field and $\mathbf{x}'(s) = \mathrm{d}\mathbf{x}/\mathrm{d}s$ is the derivative of signal $\mathbf{x}$ w.r.t. time $s$, thus "$\mathbf{F}_\theta(\mathbf{z}(s))\mathbf{x}'(s)$" is a matrix-vector multiplication. During implementation, Kidger et al. (2020) argued that a simple cubic spline interpolation on $\mathbf{x}$ allows dense calculation of $\mathbf{x}'(t)$ at any time $t$, and exhibits promising performance. In Kidger et al. (2020), it is also mathematically shown that, incorporating observations/controls following Eq. (3) is with greater representation ability compared to Eq. (2), hence achieving state-of-the-art performance on several public tasks.

**Self-attention.** It is argued in Lu et al. (2020); Vuckovic et al. (2020) that a basic unit in Transformer (Vaswani et al., 2017) with self-link consisting of one self-attention layer and point-wise feedforward layer amounts to simulating a multi-particle dynamical system. Considering such a layer with $n$ attention-heads (corresponding to $n$ particles), given an attention head index $i \in \{1, 2, ..., n\}$, the

update rule of the $i$th unit at depth $l$ reads:

$$\tilde{\mathbf{z}}_{l,i} = \mathbf{z}_{l,i} + \text{MultiHeadAtt}_{W_{\text{att}}^l}\left(\mathbf{z}_{l,i}, [\mathbf{z}_{l,1}, ..., \mathbf{z}_{l,n}]\right) \tag{4a}$$

$$\mathbf{z}_{l+1,i} = \tilde{\mathbf{z}}_{l,i} + \text{FFN}_{W_{\text{ffn}}^l}\left(\tilde{\mathbf{z}}_{l,i}\right) \tag{4b}$$

where $\text{MultiHeadAtt}_{W_{\text{att}}^l}$ and $\text{FFN}_{W_{\text{ffn}}^l}$ are multi-head attention layer and feedforward layer with parameters $W_{\text{att}}^l$ and $W_{\text{ffn}}^l$, respectively. Eq. (4) can then be interpreted as an interacting multi-particle system:

$$\frac{\mathrm{d}\mathbf{z}_i(t)}{\mathrm{d}t} = F(\mathbf{z}_i(t), [\mathbf{z}_1(t), ..., \mathbf{z}_n(t)], t) + G(\mathbf{z}_i(t)) \tag{5}$$

where function $F$ corresponding to Eq. (4a) represents the diffusion term and $G$ corresponding to Eq. (4b) stands for the convection term. Readers are referred to Lu et al. (2020); Vuckovic et al. (2020) for more details.

**Projected DEs.** It is a tool depicting the behavior of dynamics where solutions are constrained within a set. Concretely, given a closed polyhedral $\mathcal{K} \subset \mathbb{R}^n$ and a mapping $H : \mathcal{K} \to \mathbb{R}^n$, we can introduce an operator $\Pi_{\mathcal{K}} : \mathbb{R}^n \times \mathcal{K} \to \mathbb{R}^n$ which is defined by means of directional derivatives as:

$$\Pi_{\mathcal{K}}(\mathbf{a}, H(\mathbf{a})) = \lim_{\alpha \to 0_+} \frac{P_{\mathcal{K}}(\mathbf{a} + \alpha H(\mathbf{a})) - \mathbf{a}}{\alpha} \tag{6}$$

where $P_{\mathcal{K}}(\cdot)$ is a projection onto $\mathcal{K}$ in terms of Euclidean distance:

$$\|P_{\mathcal{K}}(\mathbf{a}) - \mathbf{a}\|_2 = \inf_{\mathbf{y} \in \mathcal{K}} \|\mathbf{y} - \mathbf{a}\|_2 \tag{7}$$

Intuitively, Eq. (6) pictures the dynamics of $\mathbf{a}$ driven by function $H$, but constrained within $\mathcal{K}$. Whenever $\mathbf{a}$ reaches beyond $\mathcal{K}$, it would be projected back using Eq. (7). By extending Eq. (6), Dupuis & Nagurney (1993); Zhang & Nagurney (1995) considered the projected differential equations as follows:

$$\frac{\mathrm{d}\mathbf{a}(t)}{\mathrm{d}t} = \Pi_{\mathcal{K}}(\mathbf{a}, H(\mathbf{a})) \tag{8}$$

which allows for discontinuous dynamics on $\mathbf{a}$.

## 3.2 Learning to decouple

Our method is built upon the assumption that cluttered sequential observations are composed from several relatively independent sub-systems, and therefore explicitly learns to decouple them as well as to capture the mutual interactions with a meta-system in parallel. Let the cluttered observations/controlls be $\mathbf{c}(t) \in \mathbb{R}^k$ at time $t$ for $t = 1, ..., T$, where $T$ is the time horizon. We employ $k$ distinct mappings with learnable parameters (e.g., MLP) to obtain respective controls to each sub-system: $\mathbf{x}_i(t) = p_i(\mathbf{c}(t)) \in \mathbb{R}^m$ for $i = 1, ..., n$. A generic dynamics of the proposed method can be written as:

$$\frac{\mathrm{d}\mathbf{z}_i(t)}{\mathrm{d}t} = f_i\left(\mathbf{z}_i(t), [\mathbf{z}_1(t), ..., \mathbf{z}_n(t)], \mathbf{x}_i(t), \mathbf{a}(t)\right) \quad \text{for } i = 1, ..., n \tag{9a}$$

$$\frac{\mathrm{d}\mathbf{a}(t)}{\mathrm{d}t} = \Pi_{\mathcal{S}}\left(\mathbf{a}(t), g(\mathbf{a}(t), [\mathbf{z}_1(t), ..., \mathbf{z}_n(t)])\right) \tag{9b}$$

where Eq. (9a) and Eq. (9b) refer to the $i$th sub-system describing the evolution of a single latent entity and meta-system depicting the interactions, respectively. $\mathbf{z}_i(t) \in \mathbb{R}^q$ is the hidden state for the $i$th subsystem, and $\mathbf{a}$ is a tensor governs the dynamics of the interactions. Here $\Pi_{\mathcal{S}}(\cdot)$ is a projection operator, which projects the evolving trajectory into set $\mathcal{S}$. We introduce such a operator as it is assumed that interactions among latent entities should be constrained following some latent manifold structure. $f_i(\cdot)$ and $g(\cdot)$ are both learnable functions, and also the essential roles for capturing the underlying complex dynamics.

According to Eq. (9), we fully decouple a complex system into several components. Although we found some decoupling counterparts in the context of RNNs (Li et al., 2018; Yu et al., 2020) and attention-like mechanism (Lu et al., 2020; Goyal et al., 2021), their decoupling could not be applied to our problem. We elaborate on the details of implementing Eq. (9) in the followings.

**Learning sub-systems.** Sub-systems corresponding to the latent entities seek to model relatively independent dynamics separately. Specifically, we employ the way of integrating $\mathbf{x}_i$s into Eq. (9a) in a controlled dynamical fashion as in the state-of-the-art method (Kidger et al., 2020):

$$d\mathbf{z}_i(t) = \mathbf{F}_i\left(\mathbf{z}_i(t), \mathbf{a}(t), [\mathbf{z}_i(t), ..., \mathbf{z}_n(t)]\right) d\mathbf{x}_i(t) \tag{10}$$

where $\mathbf{F}_i(\cdot) \in \mathbb{R}^{q \times m}$ is a learnable vector field. Concretely, if we let $\mathbf{Z}(t) = [\mathbf{z}_i(t), ..., \mathbf{z}_n(t)]$ be the tensor collecting all sub-systems, the $i$th sub-system in a self-attention fashion reads:

$$d\mathbf{z}_i(t) = \mathbf{F}([\mathbf{A}(t) \cdot \mathbf{Z}(t)]_i)d\mathbf{x}_i(t) \tag{11}$$

where $[\cdot]_i$ takes the $i$th slice from a tensor. Note timestamp $t$ can be arbitrary, resulting in an irregularly sampled sequential data. To address this, we following the strategy in Kidger et al. (2020) by performing cubic spline interpolation on $\mathbf{x}_i$ over observed timestamp $t$, resulting in $\mathbf{x}_i(t)$ at dense time $t$. Note that for all sub-systems, different from Eq. (10) we utilize an identical function/network $\mathbf{F}(\cdot)$ as in Eq. (11), but with different control sequence $\mathbf{x}_i(t) = p_i(\mathbf{c}(t))$. Since in our implementation $p_i(\cdot)$ is a lite network such as MLP, this can significantly reduce the parameter size.

**Learning interactions.** In our approach, interactions between latent entities are modeled separately as another meta-system. This is quite different from some related methods (Lu et al., 2020; Vuckovic et al., 2020) where sub-systems and interactions are treated as one holistic step of forward integration. Such a thorough decoupling also brings about gradient with higher magnitude in the early training stage (see Figure 2), which allows for faster convergence and more significant gradient backpropagation. For the meta-system describing the interactions in Eq. (9b), two essential components are involved: domain $\mathcal{S}$ and the projection operator $\Pi$. In the context of ProjDEs, a system is constrained as $\mathbf{a}(t) \in \mathcal{S}$ for any $t$. In terms of interactions, a common choice of $\mathcal{S}$ is the stochastic simplex which can be interpreted as a transition kernel (Vuckovic et al., 2020). We allow follow this setting by defining $\mathcal{S}$ be a row-wise stochastic $(n-1)$-simplices:

$$\mathcal{S} \triangleq \{\mathbf{A} \in \mathbb{R}^{n \times n} | \mathbf{A}\mathbf{1} = \mathbf{1}, \mathbf{A}_{ij} \geq 0\} \tag{12}$$

where $\mathbf{1}$ is a vector with all 1 entries. $\mathbf{A} = \text{mat}(\mathbf{a})$ is a $n \times n$ matrix. In the sequel, we will use the notation $\mathbf{A}$ throughout. Thus the meta-system capturing the interactions can be implemented as:

$$\frac{d\mathbf{A}(t)}{dt} = \Pi_{\mathcal{S}}\left(\mathbf{A}(t), g(\mathbf{A}(t), [\mathbf{z}_1(t), ..., \mathbf{z}_n(t)])\right) \tag{13}$$

Notably, the projection operator $(\text{softmax}(\cdot))$ in the attention mechanism can be viewed as a row-wise projection onto the $(n-1)$-simplex with entropic regularization (Amos, 2019):

$$P_{\mathcal{S}}(\mathbf{A}_{j,:}) = \min_{\mathbf{B} \in \mathcal{S}} -\mathbf{A}_{j,:}\mathbf{B}_{:,j} - \mathbb{H}(\mathbf{B}_{:,j}) \tag{14}$$

where $\mathbb{H}(\cdot)$ is the entropy function. $\mathbf{A}_{j,:}$ and $\mathbf{B}_{:,j}$ are the $i$th row and column of $\mathbf{A}$ and $\mathbf{B}$, respectively. It is readily derived that row-wise $\text{softmax}(\mathbf{A})$ is the solution to Eq. (14) by utilizing KKT condition. Comparing Eq. (14) to the standard Euclidean projection in Eq. (7)j, we note the entropic regularization $\mathbb{H}(\cdot)$ in Eq. (14) allows for a smoothed trajectory without discontinuity, thus eases the gradient flow over the whole integration path.

While Eq. (13) defines a projected dynamical system directly on $\mathbf{A}$, we switch to update the system using $\mathbf{L}$ as follows, which is considered to further ease the forward integration. This is achieved by instead modeling the dynamics of the feature before fed into $\text{softmax}(\cdot)$:

$$\mathbf{A}(t) = \text{Softmax}(\mathbf{L}(t)) \tag{15a}$$

$$\mathbf{L}(t) = \mathbf{L}(0) + \int_0^t \frac{d}{ds} \frac{\mathbf{Q}(\mathbf{Z}(s)) \cdot \mathbf{K}^\top(\mathbf{Z}(s))}{\sqrt{d_k}} ds, \tag{15b}$$

$$\mathbf{L}(t + \Delta t) = \mathbf{L}(t) + \Delta t \cdot \frac{d}{ds} \frac{\mathbf{Q}(\mathbf{Z}(s)) \cdot \mathbf{K}^\top(\mathbf{Z}(s))}{\sqrt{d_k}}\bigg|_{s=t} \tag{15c}$$

where $\mathbf{Q}(\cdot)$ and $\mathbf{K}(\cdot)$ correspond to the query and key in the attention mechanism, respectively. $\mathbf{L}(0) = \mathbf{Q}(\mathbf{Z}(0)) \cdot \mathbf{K}^\top(\mathbf{Z}(0))/\sqrt{d_k}$. We show that updating the dynamic of $\mathbf{L}$ following Eq. (15) is equivalent to directly updating $\mathbf{A}$ in Appendix A.2.

We employ the standard Euler's descritization for performing the integration, by updating $\mathbf{z}$ and $\mathbf{A}$ simultaneously. We term our approach decoupling-based neural system (**DNS**).

## 4 EXPERIMENTS

We evaluate the performance of DNS on multiple synthetic and real-world datasets. More details about the dataset and implemetation details can be found in Appendix A.9. Throughout all the tables consisting of the results, a placeholder "✗" is referred to a failure during training process after multiple trials, and "-" indicates "not applicable" (RIM cannot handle irregular cases).

**Baselines.** We compare DNS with several selected models capturing interactions or modeling irregular time series, including **CT-RNN** (Funahashi & Nakamura, 1993) modeling continuous dynamics, **CT-GRU** (Mozer et al., 2017) using state-decay decay mechanisms, **RIM** (Goyal et al., 2021) updating almost independent modules discretely, and **NeuralCDE** (Kidger et al., 2020) which reports state-of-the-art performance on several benchmarks. We use "method $x$ $y$" and "method $y$" to indicate settings, where $x$ and $y$ are the number of the underlying modules (e.g., sub-systems in our method and independent components in RIM) and dimension of the latent feature.

Table 1: **Trajectory prediction**. Three body.

| Model | Square-Error ($\times 10^{-2}$) | |
| --- | --- | --- |
| | Regular | Irregular |
| CT-RNN 512 | 2.1291 | 3.0538 |
| CT-RNN 1024 | 1.9943 | 2.8968 |
| CT-RNN 2048 | 1.9514 | 3.0008 |
| CT-GRU 512 | 1.8554 | 2.4997 |
| CT-GRU 1024 | 1.7155 | 2.3020 |
| CT-GRU 2048 | 1.6421 | 2.1996 |
| NeuralCDE 512 | 3.9213 | 5.0077 |
| NeuralCDE 1024 | 4.4934 | 5.4811 |
| RIM 512 | 2.4519 | - |
| RIM 1024 | 2.1865 | - |
| DNS 512 | 1.8272 | 2.4874 |
| DNS 1024 | 1.8206 | 2.3650 |
| DNS 2048 | **1.5012** | **2.0994** |

Table 2: **Link predition.** Charged.

| Model | Accuracy(%) | |
| --- | --- | --- |
| | Regular | Irregular |
| CT-RNN 64 | 61.86 | 61.90 |
| CT-RNN 128 | 62.60 | 63.47 |
| CT-RNN 256 | 62.54 | 65.40 |
| CT-GRU 64 | 63.13 | 63.23 |
| CT-GRU 128 | 63.76 | 63.64 |
| CT-GRU 256 | 64.18 | 65.16 |
| NeuralCDE 64 | 67.90 | 65.83 |
| NeuralCDE 128 | 68.92 | **66.97** |
| RIM 64 | 66.46 | - |
| RIM 128 | **69.14** | - |
| DNS 5 64 | 67.27 | 66.04 |
| DNS 5 128 | 69.07 | 65.23 |
| DNS 5 256 | 68.37 | 64.99 |

### 4.1 THREE BODY

Three body problem is characterized by a chaotic dynamical system for most randomly initial conditions. A small perturbation may cause drastic changes of the movement. Taking into account the problem complexity, it is particularly suitable for testing our approach. In this experiment, we consider a trajectory predicting problem given the noisy historical motion of three masses, where gravity causes interactions between them. Therefore, models need to (implicitly) learn both Newton's laws of motion for modeling sub-system dynamics and Newton's law of universal gravitation to decouple the latent interaction. This dataset consists of 50k training samples and 5k test samples. For each sample, 8 historical locations for regular setting and 6 historical locations (randomly sampled from 8) for irregular setting in 3-dimensional space of three bodies are given to predict 3 subsequent locations, hence a 9-dimensional observation at each time stamp. To equip with the cluttered setting, the correspondence between dimensions and bodies will not be fed into the learning models. Each model is trained until convergence and the average of mean squared error (MSE) of the last 5 epochs on the test set is reported as the accuracy. Results are summarized in Table 1. We can conclude that "DNS 2048" outperformed all the selected counterparts in both regular and irregular settings. Notably, although our method is built on NeuralCDE, but with the decoupling, the performance can be significantly improved.

**Latent entity V.S. physical entity.** We visualize dynamics **A** along the movements of three body system. Note we set up 3 sub-systems for potentially capturing dynamics of 3 physical entities. Figure 1 shows that when three bodies approach each other, the off-diagonal entries of **A** increase, indicating stronger interactions when physical bodies are close to each other. This implies that DNS can somewhat decouple a complex three body system according to its physical meaning in the real world.

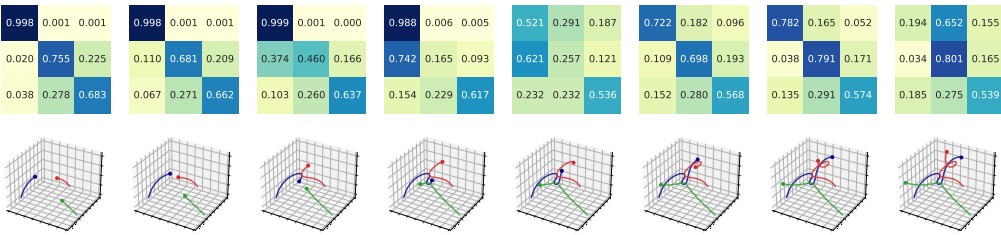

Figure 1: A figure showing the evolution over time on interactions (the upper row) between three **latent sub-systems**, as well as the corresponding three **physical bodies** in a Three Body environment.

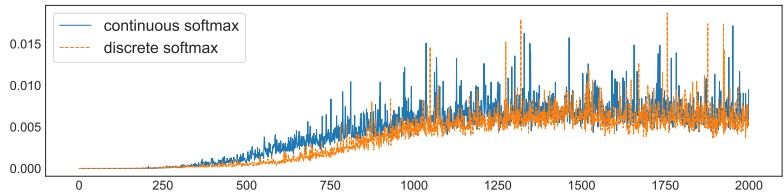

Figure 2: Comparison of magnitude of gradient of $\mathbf{A}$ along with the training process. "Continuous softmax" and "Discrete softmax" correspond to the update rules following Eq. (15) and holistic forward integration as in Lu et al. (2020); Vuckovic et al. (2020), respectively.

## 4.2 SPRING

We experiment the capability of DNS in decoupling the independence in complex dynamics controlled by simple physics rule. We use a simulated system in which particles are connected by (invisible) springs (Kuramoto, 1975; Kipf et al., 2018). Each pair of particles has equal probability of having interaction or not. Our task is to use observed trajectory to predict whether there are springs between any pair of two particles, which is analogous to the task of link prediction under a dynamical setting. This can be inferred from whether two trajectories change coherently. Spring dataset consists of 50k training examples and 10k test examples. Each sample has the length of 49. We also test model's noise resistance and add Gaussian noise to the observed data. We set up two scenarios "Train&Test" and "Test" (see Table 3) corresponding to injecting noise at both training and testing phases, and only at testing phases, respectively. We test a variety of combinations of the number of sub-systems and dimension of hidden state. Experimental results are in Table 3. From the first two blocks of Table 3, we see standard DNS stably outperforms all the selected counterparts by a large margin when the data is clean. When noise exists on both training and testing phases, DNS exhibits somewhat sensitivity to the coefficients, thus needs to be fine-tuned. However, when only testing samples are noisy, DNS again maintains high stability and promising performance.

**Variants of DNS.** To examine the compatibility of our model, we create two variants of DNS. The first version is to replace cubic spline interpolation over $\mathbf{x}_i(t)$ with natural smoothing spline (Green & Silverman, 1993), in consideration of incorporating smoother controls. This version is termed as $\mathrm{DNS_S}$. Another version of DNS is without explicitly modeling the dynamics of $\mathbf{A}$ as in Eq. (15), but treating one step of integration as a holistic forward pass consisting of both $\mathbf{z}$ and $\mathbf{A}$ following the interpretation in Lu et al. (2020); Vuckovic et al. (2020). This variant is called $\mathrm{DNS_D}$. Evaluation results on these two variants are summarized in the third block in Table 3. According to Table 3, even though $\mathrm{DNS_S}$ does not exhibit the best performance with clear data, it is quite reliable in the noisy cases. It seems a smoothing procedure on the controls can be helpful under massive uncertainty. This also raises one of our future research directions to investigate the way handling different controls. $\mathrm{DNS_D}$, with a holistic forward integration, cannot achieve performance comparative to standard DNS. To analyze, we visualize the magnitude of gradient of $\mathbf{A}$ (with DNS and $\mathrm{DNS_D}$) along with the training process as in Figure 2. One can readily find that the proposed method DNS by fully decoupling $\mathbf{A}$ can derive a more significant gradient in the early training stage than $\mathrm{DNS_D}$. We argue that, explicitly introducing interaction dynamics $\mathbf{A}$ creates a "highway" in parallel to each

Table 3: **Link prediction**. Spring dataset.

| Model | Clean | | Noisy | | Param Size(MB) |
|---|---|---|---|---|---|
| | Regular | Irregular | Train&Test | Test | |
| CT-RNN 128 | 71.39 | 76.96 | 70.62 | 69.91 | 0.1 |
| CT-RNN 256 | 76.68 | 81.00 | 75.88 | 75.99 | 0.3 |
| CT-RNN 512 | 72.44 | 51.68 | 75.09 | 71.46 | 1.1 |
| CT-GRU 128 | 86.20 | 87.13 | 85.90 | 86.14 | 1.3 |
| CT-GRU 256 | 88.14 | 87.92 | 88.13 | 88.11 | 4.8 |
| CT-GRU 512 | 89.57 | 88.61 | 89.17 | 89.24 | 18.0 |
| CT-GRU 1024 | 90.97 | 89.99 | 90.25 | 90.84 | 70.0 |
| NeuralCDE 128 | 90.02 | 89.55 | 77.32 | 75.51 | 1.3 |
| NeuralCDE 256 | 91.06 | 90.55 | 80.12 | 77.79 | 5.3 |
| NeuralCDE 512 | 91.57 | 91.40 | 81.87 | 78.22 | 21.1 |
| NeuralCDE 1024 | 91.23 | 90.71 | 80.47 | 79.15 | 84.2 |
| RIM 128 | 85.77 | - | 85.82 | 85.74 | 9.9 |
| RIM 256 | 86.92 | - | 86.64 | 86.91 | 25.7 |
| RIM 512 | 87.43 | - | 87.42 | 87.43 | 75.4 |
| DNS 5 128 | 93.29 | 93.15 | 91.96 | 87.97 | 2.8 |
| DNS 5 256 | 93.81 | 93.43 | 92.56 | 86.36 | 10.7 |
| DNS 5 512 | 88.07 | 88.04 | 89.74 | 83.64 | 41.8 |
| DNS 8 128 | 93.59 | 94.43 | 66.72 | 85.40 | 2.9 |
| DNS 8 256 | 94.92 | **95.62** | 93.73 | 87.24 | 11.0 |
| DNS 8 512 | **95.38** | 95.40 | 76.52 | 89.44 | 42.3 |
| DNS 10 128 | 93.79 | 93.04 | 93.11 | 86.00 | 3.1 |
| DNS 10 256 | 94.86 | 95.20 | 93.90 | 88.17 | 11.2 |
| DNS 10 512 | 92.99 | 92.79 | 91.48 | 87.41 | 42.7 |
| $DNS_S$ 128 | 93.00 (5) | 93.15 (5) | 93.73 (8) | 91.82 (8) | 3.1 |
| $DNS_S$ 256 | 93.99 (5) | 93.43 (5) | **94.15** (8) | 78.65 (8) | 11.2 |
| $DNS_S$ 512 | 95.04 (5) | 88.04 (5) | **94.15** (8) | **93.26** (8) | 42.7 |
| $DNS_D$ 5 128 | 92.91 | 93.28 | 91.40 | 87.61 | 2.8 |
| $DNS_D$ 5 256 | 93.80 | 94.00 | 92.22 | 87.16 | 10.7 |
| $DNS_D$ 5 512 | 88.78 | 91.93 | 89.65 | 81.92 | 41.8 |

sub-system, which tends to ease the gradient flow throughout the training, and is thus beneficial for training on complex and cluttered sequential data.

### 4.3 CHARGED

Experiment setting of charged dataset follows Kuramoto (1975); Kipf et al. (2018). Charged particles attract or repel each other with equal probability. Interaction strength is inversely proportional to the square of the distance and results a more complex system compared with spring dataset. Particles are more frequently bounced back by the boundary of the container, leading to a sudden change of trajectory. Hence, intuitively, models introduced for continuous dynamics may have limited performance. Experiment results are in Table 2. In this dataset, DNS achieved promising performance close to RIM and NeuralCDE. We observe continuous-time versions of RNNs (CT-RNN and CT-GRU) don't perform well under such setting with several changes at boundary, yet DNS achieves a good accuracy.

### 4.4 HUMAN ACTIONS

Recognition of human actions dataset contains three types of human actions which are hand clapping, hand waving and jogging (Schuldt et al., 2004). We take 50 equispaced frames from each video. For this dataset, we consider limbs of the character as subsystems. When the character does one kind of actions, subsystems interact in a specific pattern. We test the compatibility of all the selected models by turning the parameters of the backbone Resnet18 (He et al., 2016) untunable and tunable.

Table 4: **Video classification**. Regular and Irregular Human Actions.

| Model | Unnormalized | | | Normalized | |
|---|---|---|---|---|---|
| | Regular | | Irregular | Regular | Irregular |
| | Frozen | Trainable | Trainable | Frozen | Trainable |
| CT-RNN 32 | 33.33 | 46.17 | 48.00 | 88.33 | 37.22 |
| CT-RNN 64 | 33.50 | 47.50 | 58.33 | 90.00 | 24.44 |
| CT-GRU 32 | 63.83 | 58.33 | 56.00 | **96.67** | 66.67 |
| CT-GRU 64 | 37.00 | 60.33 | 66.67 | **96.67** | 72.78 |
| NeuralCDE 32 | 68.33 | 52.47 | 57.83 | 81.67 | 33.89 |
| NeuralCDE 64 | 70.50 | 70.33 | 59.17 | $\times$ | 61.67 |
| RIM 32 | 49.33 | 55.50 | - | 95.00 | - |
| RIM 64 | 43.00 | 44.83 | - | 95.00 | - |
| DNS 32 | 70.17 | 95.00 | **95.33** | **96.67** | 86.67 |
| DNS 64 | **78.67** | **97.00** | 93.17 | **96.67** | **87.78** |

We also feed the model with data in raw format (pixel value in each channel is in $[0, 255]$) and normalized format (pixel value in each channel follows Gaussian $\mathcal{N}(0.5, 0.5)$), to examine robustness of different methods under various changing scales. Experimental results are summarized in Table 4. In unnormalized case, DNS significantly outperformed all other counterparts, demonstrating strong robustness for raw data collected from a large range. DNS is also more compatible to backbone of Resnet18, and thus potentially being more flexible to be integrated into various tasks. In the normalized setting, although DNS and CT-GRU hold comparative accuracy, DNS delivers prominent performance under the irregular setting.

To view how the decoupling works for video recognition task, we visualize the strength of the learned parameters mapping a 128-D feature into 6 latent sub-systems in Figure 3. For better view, we re-organize the order of the 128-D feature. It can be seen that there are some latent structure in the parameters grouping 128-D control to the system. Each sub-system mainly focuses on a small portion of the control, based on which we can infer that each sub-system models different components in inputted images.

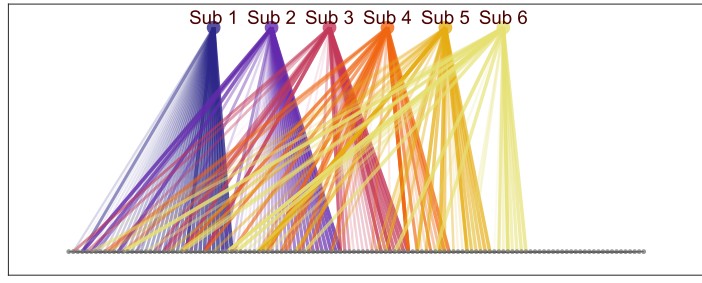

Figure 3: A figure showing the importance of each feature vector entry for subsystems

## 5 CONCLUSION

In this paper, we propose a method for modeling cluttered and irregularly sampled sequential data. Our method is built upon the assumption that complex observation may be derived from relatively simple and independent latent sub-systems, wherein the interactions also evolve over time. We devise a strategy to explicitly decouple such latent sub-systems and a meta-system governing the interaction. Inspired by recent findings about self-attention mechanism, and further utilizing the tool of projected differential equations, we present an novel interpretation to our model. Experiments on various tasks demonstrate prominent performance of our method over previous state-of-the-art methods.

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

# A APPENDIX

## A.1 DETAILS ABOUT FIGURE 3.

We replace the fully connected layer in the pretained Resnet18 with another neural network whose output size equals to 64. Freezing Resnet means that we do not update parameters in the convolutional layer. Parameters of the fully connected layers can still be updated. Image feature vectors are fed forward by a linear layer of size 64 by 128 and activated by ReLu function. Then, feature vectors are fed forward by distinct linear layer and we obtain different control signal for each subsystem. In Figure 3, gray points on the second line denotes entries of the 128 dimensional feature vector after reordering. For each subsystem, we plot the top 40 entries which has the greatest impact for control signals.

When plotting the figure of gradients change along the training process, we drop the points where gradients are relatively too large. In practice, we can avoid gradient exploding gradient problem by gradient clipping.

## A.2 ON THE EQUIVALENCE OF MODELING $\frac{d\mathbf{A}}{dt}$ AND $\frac{d\mathbf{L}}{dt}$

Let $\mathbf{L}(t)$ denotes the multiplication of key and query, i.e., $\mathbf{L}(t) = \frac{\mathbf{Q}(t)\mathbf{K}^\top(t)}{\sqrt{d_k}}$, the $\mathbf{A} = \mathrm{softmax}(\mathbf{L})$. If we model the dynamics of $\mathbf{L}(t)$, we obtain

$$\mathbf{L}(t + \Delta t) = \mathbf{L}(t) + \Delta t \cdot \frac{d\mathbf{L}}{dt}, \tag{16}$$

Apply the $\mathrm{softmax}$ function on both side of the equation, we have

$$\mathbf{A}(t + \Delta t) = \mathrm{softmax}(\mathbf{L}(t) + \Delta t \cdot \frac{d\mathbf{L}}{dt}) + \mathbf{A}(t) - \mathbf{A}(t)$$

$$= \mathbf{A}(t) + \mathrm{softmax}(\mathbf{L}(t) + \Delta t \cdot \frac{d\mathbf{L}}{dt}) - \mathrm{softmax}(\mathbf{L}(t))$$

Reorder the equation, we have

$$\frac{\mathbf{A}(t + \Delta t) - \mathbf{A}(t)}{\Delta t} = \frac{\mathrm{softmax}(\mathbf{L}(t) + \Delta t \cdot \frac{d\mathbf{L}}{dt}) - \mathrm{softmax}(\mathbf{L}(t))}{\Delta t}$$

$$= \frac{\mathrm{softmax}(\mathbf{L}(t) + \Delta t \cdot \frac{d\mathbf{L}}{dt}) - \mathrm{softmax}(\mathbf{L}(t))}{\Delta t \cdot \frac{d\mathbf{L}}{dt}} \cdot \frac{d\mathbf{L}}{dt}$$

Take $\Delta t \to 0$, we have

$$\frac{d\mathbf{A}}{dt} = \frac{d\mathrm{softmax}(\mathbf{L}(t))}{d\mathbf{L}} \cdot \frac{d\mathbf{L}}{dt}, \tag{17}$$

which is equivalent to the update step in Eq. (15).

## A.3 EXPERIMENT DETAILS

### A.3.1 DNS

For the implementation simplicity, DNS with batch input requires each sample be observed at the fist and last timestamp. Default control signal dimension equals to $2\times$ input_size. When initilizing the Weight matrix of key and query layer, control encoder and initial hidden state encoder, we use $0.01\times$ torch.rand and set bias equals to 0.

## A.4 CT-RNN

We set the ODE solver to Euler with 3 folds for each integration, and we set $\tau = 1$.

## A.5 CT-GRU

We set model's hyperparameters as $M = 8, \tau = 1$.

### A.6 NEURALCDE

We use a two layer neural network to approximate ODE function. We use Euler method to solve the CDE on three body to accelerate training process. We use torchdiffeq.odeint_adjoint library and set the numerical precision to 0.01 to solve the CDE on other datasets.

### A.7 RIM

We set the hyperparameters to the default values in the orginal paper. Key size input :64, value size input: 400, query size input 64, number of input heads: 1, number of common heads: 4, input dropout: 0.1, common dropout: 0.1, key size common: 32, value size common: 100, query size common: 32.

Number of units and k: Spring: (8, 5), Charged: (8, 5), Three body: (6, 3), Human action: (6, 3).

### A.8 TRAINING HYPERPARAMETERS

Table 5 shows the default hyperparameters selection to train each model. For three body dataset, we find that for all models, setting batch size to be 1 can enhance model's performance.

|  | three body | irregular spring/charged | human action |
|---|---|---|---|
| minibatch size | 1 | 128 | regular: 4, irregular: 1 |
| optimizer | Adam | | |
| learning rate | 1e-3 using a cosine annealing schedule with eta_min=5e-5 | | |
| clip grad max norm | 0.1 | × | 0.1 |
| epochs | 100 | regular: 25, irregular: 50 | 50 |

Table 5: Training Hyperparameters

### A.9 DATASET SETTINGS

#### A.9.1 THREE BODY DATASET

We use Python to simulate the motion of three bodies. We add a multiplication noise from a uniform distribution $\mathcal{U}(0.995, 1.005)$. We generate 50k training samples and 5k test samples. Three celestial bodies in all samples have a fixed initial position and each pair has the sample distance. We randomly initialize the velocity so that in most samples, all three bodies have strong interactions and it is also possible that only two celestial bodies have a strong interaction and the rest one moves almost in a straight line. The dataset contains the locations of three bodies in three dimensional space, so the input size equals to 9. All samples have the length of 8. For partially observed dataset, all samples have the length of 6 and the locations at the last timestamp are always observed. We set batch size equals to 1 and update models after 128 forward call. We use the historical motion of three bodies to predict 3 subsequent motion. We grid search the hidden size in [512, 1024, 2048] and use the mean value of mean squared error (MSE) of the last 5 epochs on the test as the model accuracy.

#### A.9.2 SPRING/CHARGED

We follow the experiment setting in Kipf et al. (2018). We generate 40k training samples and 10k test samples. We test model's noise resistance ability on noisy spring dataset. The noise level can be see from Figure 4. We set the number of particles to be 5. The input contains current location and velocity of each particle in two dimension, so the input size is 20. All samples have the length of 49/19 (spring regular/irregular) and 24/9 (charged regular/irregular). Features at the first and last timestamp are always observed. We set batch size equals to 128. The task is to prediction whether there are springs connecting two particles. We grid search the hidden size in the range of 128, 256, 512 for DNS and RIM and in the range of 128, 256, 512, 1024 for CTGRU and NeuralCDE. We use the mean accuracy of the last 5 epochs on test set as the model accuracy (early stop if the mean accuracy of the current epoch is 5% (regular) or 2% (irregular) less than the last epoch).

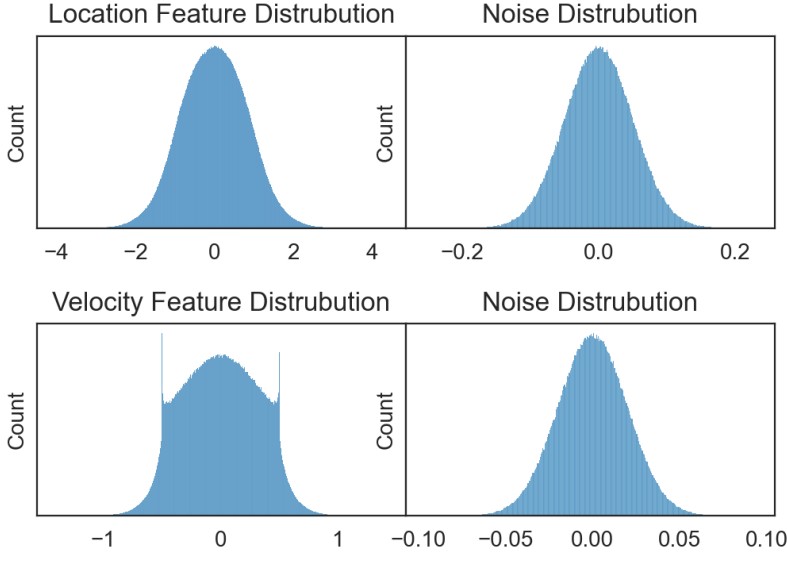

Figure 4: noise level

### A.9.3 HUMAN ACTION

Human action dataset contains three types of human actions. There are 99 videos for hand clapping, 100 videos for hand waving and 100 videos for jogging. Videos have the length of 15 seconds in average and all videos were taken over homogeneous backgrounds with a static camera. We downsample the resolution of each frame to 224×224 pixels. We use mean and std equals to 0.5 to normalize images and use Resnet18 pretained on ImageNet (He et al., 2016) as feature extractors. For regular human action dataset, we set batch size equals to 4. Because the classification task is relatively easy, we froze the convolutional layer parameters in Resnet18 and set the output size of the fully connected layer after the convolutional layer to 64. Models need to use clustered image features for action recognition. For irregular human action dataset, each video has the length of 36 to 50 frames and we set batch size equals to 1. We fine tune Resnet18 and also set the dimension of image feature vector to 64. We grid search the hidden size in [32, 64] and use the mean accuracy of the last 3 epochs on test set as the model accuracy.

## B CROSS VALIDATION

Table 6: **Link prediction**. Spring dataset with cross-validation.

| Model | Accuracy (%) | |
| --- | --- | --- |
| | Regular | Irregular |
| CT-RNN 256 | $74.583 \pm 2.421$ | $53.262 \pm 1.013$ |
| CT-GRU 1024 | $90.362 \pm 0.175$ | $89.791 \pm 0.133$ |
| NeuralCDE 256 | $91.076 \pm 0.121$ | $90.521 \pm 0.107$ |
| RIM 512 | $87.450 \pm 0.083$ | - |
| DNS 512 | $\mathbf{94.755 \pm 0.584}$ | $\mathbf{93.695 \pm 2.142}$ |

We use 5-fold cross-validation, leaving the test set unseen during the training process. We early stop the training process if the accuracy on the training set of the current epoch is 3% less than the previous best epoch. Results are summarized in Table 6 with the mean accuracy and the corresponding standard deviation. We can readily see that DNS outperforms all the other methods by a large margin.

## C ABLATION STUDY

Since our method merely incorporates an extra meta-system and a control encoder for modeling the interaction compared to standard NeuralCDE, we conduct experiments under different settings to see how different encoders and hidden state dimension can contribute to improve NeuralCDE. To ensure the fairness, we cast a 2-layer MLP with different output sizes (2 and 16 times of input size) as in DNS to obtain the feature of controls, with varying sizes of the hidden state (128, 256 and 512). Results are summarized in Table 7. Note "DNS 8 512" incorporates 8 MLPs, which is much less than the "MLP (output=16×input size)" in terms of parameter size. We see that with an extra control encoder and varying hidden dimensions, there is no obvious performance difference among these settings. However, once the interaction meta-system is imposed, DNS can achieve quite significant performance gain. This in turn shows the necessity of the proposed meta-system for explicitly modeling the evolving interactions.

Table 7: Ablation study of DNS

| Control | Accuracy |
|---|---|
| No encoding + 128 | 90.0209 |
| No encoding + 256 | 91.0624 |
| No encoding + 512 | 91.5740 |
| MLP(output=2×input size) + 128 | 87.0086 |
| MLP(output=2×input size) + 256 | 90.8679 |
| MLP(output=2×input size) + 512 | 91.5120 |
| MLP(output=16×input size) + 128 | 91.1651 |
| MLP(output=16×input size) + 256 | 91.0800 |
| MLP(output=16×input size) + 512 | 90.7029 |
| DNS 8 512 (8×MLP(output=2×input size)) + 512 | 95.38 |

