# OpenReview forum: "Learning to Decouple Complex System for Sequential Data"
_ICLR.cc/2023/Conference — Submitted to ICLR 2023_

### Official Review · Reviewer_gYdf · 2022-10-23

**Confidence:** 4
**Correctness:** 4
**Technical Novelty And Significance:** 4
**Empirical Novelty And Significance:** 3
**Recommendation:** 8

**Clarity, Quality, Novelty And Reproducibility:**

Barring the fair number of typos, the paper is really well articulated and has a lot of interesting ideas to build future work on. The code was fairly easy to follow as well and would be super useful to reproducing the experiments.

**Strength And Weaknesses:**

**Strengths**

The problem of simplifying a complex system into appropriate simpler sub-systems is one of the most important things scientists and engineers do in their daily life. Figuring out ways to automate this process directly from data can be extremely impactful. The paper very clearly formalizes the problems and identifies connections and bridges to theory of differential equations as well as continuous time interpretations of extremely successful attention mechanisms. Insights from these connections result in a very natural way of decoupling the sub-systems as well as the meta-interaction system. Results are validated on a variety of problems including human-action recognition with quite a few relevant baselines which are all appropriately modeling the task with an underlying continuous dynamics.

**Weaknesses**

The paper doesn't mention this but training neural diffeq approaches can be quite inefficient, especially on modern heterogenous architectures.
Would be great to have limitations explicated somewhere.
There are a lot of typos in the paper. There are also quite a few w.r.t `\citep` vs `\citet` issues too. Reading with fresh eyes would help fix a lot of these issues.

**Summary Of The Paper:**

The paper focuses on the problem of modeling data coming from a complex system with an underlying continuous dynamics. The key assumption/claim of the paper is that such a complex system is best modeled as an interaction between multiple coupled sub-systems (that has its own dynamics too). Using ideas from differential equation literature and modern attention mechanism, the paper comes up with a compelling approach towards modeling such systems. Advantages of the approach are the clear in the experimental results where the approach is compared against a variety of baselines.

**Summary Of The Review:**

This is a fantastic paper and likely has a wider real world impact capability especially when modeling real world man-made physical systems which are often modular. It would be great if the paper could discuss its own limitations more carefully (especially neural DiffEq can be hard to train with increasing dimensions), any effect of diffeq solver vs Euler as used here, GPUs vs CPUs, etc. W.r.t video classification, event camera data might be ideal demonstration for irregular data. For a lot of real world applications the modular structure is partially known as well and incorporating that information would be fairly impactful future work. Similarly a good test of decoupling could be if the components could be transferred to a similar but different system.

---

> ### Author Response · Authors · 2022-11-17
> **response to reviewer 4**
>
> We thank reviewer 4 for his/her constructive advice. It is also appreciated that you identify the novelty of our work. We have revised our manuscript following your suggestion.

---

### Official Review · Reviewer_gG1Z · 2022-10-24

**Confidence:** 2
**Correctness:** 2
**Technical Novelty And Significance:** 2
**Empirical Novelty And Significance:** 2
**Recommendation:** 5

**Clarity, Quality, Novelty And Reproducibility:**

The text is reasonably clear, although significant improvements can be performed. The work is of high quality deadling with challenging problems and approaches. To the best of my knowledge, this work is novel. Given the discussion above regarding evaluation, I believe that reproducibility is nearly impossible.


**Strength And Weaknesses:**

The main strength of this paper is the incorporation of two seemingly unrelated concepts: controlled differential equations and attention maps into a unified framework that models irregularly sampled time series data and their interactions. While that involved math is somewhat challenging, the authors did a good job in laying out the dependencies and relations between the various equations that appear in Sec. 3.

The main weakness of this paper is the evaluation of the proposed approach. First, the tasks chosen for evaluation seem somewhat arbitrary, and not necessarily related to the underlying assumptions. For instance, the motion of three bodies is independent as long as no interactions occur. However, the main motivation beind the coupled PDEs introduced in Sec. 3 was to model several highly independent sub-systems. In addition, the authors state that the three body problem was simulated with many interactions, so the amount of independent motion is further reduced. Second, the description of the evaluation protocol and reported results are somewhat alarming. Was there a validation set? The text suggests there was no validation set. How was grid search performed then? By reporting the best results on the (5 last epochs of) test set? If so, it might be that the results are biased towards overfitting the test sets. Moreover, the text suggest that only a single run was performed. Given that the differences in error measures are not significant, I would like to see a more extensive evaluation where testing is performed 5 or more times with different seeds, and average error metrics and standard deviations are reported. Also, are the results reported in Tabs. 1-4 reported elsewhere (for the baselines)? If not, how did achieve these results? training and testing the various methods? Was there an extensive grid search performed for the other methods? If so, how did you choose the parameter ranges? As a note, your framework uses more supervision information in comparison to other works, as you use e.g., 3 sub-systems in the three body example. Given the similarity between the proposed method and NCDE, I would like to see an extensive ablation study between these approaches. The graphs in Fig. 2 do not show any advantage to the proposed method in terms of gradient magnitudes. The issue around gradient magnitude is specifically related to exploding and vanishing gradients. Fig. 2 shows neither, and in particular, there is not even an order of magnitude difference between red and blue curves.

The second weakness is related to the modeling. Specifically, the text is dense and hard to follow at times. No proper motivation or description was provided regarding Eq. (11). Why model $dz_i(t)$ in this way? The interaction differential equation was better motivated using attention mechanisms, but still I do not necessarily see why this particular model satisfies the underlying assumptions. For instance, the interaction DE does not encourage the sub-systems to be independent. How $g$ is modeled, and what is its purpose? What is the loss function? What do you do with $z, A$ to get the output?



**Summary Of The Paper:**

This paper introduces a new neural ODE-based framework to model irregularly-sampled sequential data. The approach is based on neural control differential equations, and on the observation that attention models can be viewed as integrating a differential equation path. The authors evaluate their approach on a few toy examples, and it is compared to several other baselines (ncde included). The reported results show certain benefits to using the proposed method.


**Summary Of The Review:**

This paper starts with an intuitive requirement: one wants to model complex systems using several simple and independent sub-systems. This motivates from a higher-level the method design, which turns out to be a coupled system of differential equations, with another equation dealing with interactions. While the intuition and modeling is generally reasonable, the evaluation is lacking in terms of chosen tasks, and testing protocol. I believe the paper could be significantly stronger if the authors can improve on these points.

---

> ### Author Response · Authors · 2022-11-17
> **response to reviewer 3**
>
> We thank reviewer 3 for his/her constructive advices. Responses to some other concerns are listed as below.
>
> The assumption of our method is that a complex system may consist of several relatively independent latent sub-systems coupled by another meta-system describing the interactions. This assumption also motivated several related approaches (e.g., RIM, self-attention) which achieved promising performance on various tasks. Our approach is based on the same assumption (which was proved rational and powerful), but further motivated by an attempt to thoroughly decouple these sub-systems and a new orthogonal meta-system.
>
> In terms of the experimental settings, we set 3 sub-systems in three body task to investigate if the learned sub-systems and their interactions have any physical meaning since 3 bodies are involved. According to Fig.1, we did observe some correspondence between the latent sub-systems and the physical entities. However, we did not attempt to obtain this prior knowledge (3 physical entities). Instead, in the Spring dataset where 5 physical balls are involved, we tested the number of sub-systems 5, 8, and 10, where 8 sub-systems, not 5, achieved the best performance. This in turn indicates that our method does not need such a prior (e.g., number of physical entities) to reach a better performance.
>
> The way of modeling dz_i(t) and F() exactly follows the way in NeuralCDE, which was originally derived from Controlled Differential Equation in the context of control theory. Integrating meta-system and sub-system as [A(t) Z(t)] was inspired by self-attention, and has a deep connection to neuroscience.

---

> > ### Comment · Reviewer_gG1Z · 2022-11-17
> > **Response to authors**
> >
> > Thank you for your response.

---

### Official Review · Reviewer_koo9 · 2022-10-28

**Confidence:** 4
**Correctness:** 2
**Technical Novelty And Significance:** 2
**Empirical Novelty And Significance:** 2
**Recommendation:** 3

**Clarity, Quality, Novelty And Reproducibility:**

- Quality and novelty
    - The paper is a combination of Neural Controlled Dynamics and Transformers and, in this view, is not very novel. Furthermore, stronger results and explanations would be necessary to convince me that this combination is better than other complex architectures one may propose.
    - As mentioned under weaknesses, experimental results were sometimes much lower than published algorithms.
    - Most experimental results are missing confidence intervals and probably have too many significant digits. Furthermore, I believe most people now do not show the size of the architecture as different architectures and instead usually use the larger version (briefly explaining if validation data favored a smaller version). Not all, but most cases, shown in the tables of this paper corroborates that the bigger model tends to be better.
    - The paper says that "-" means both not training reliably or not tried. It would be important to separate these two cases, particularly as DNS has many "-" in table 3, pointing to instabilities, which would also have to be explained.
- Clarity
    - The introduction of related concepts was good, the motivation and explanation of the proposed architecture needs to be worked on in my opinion.
    - When talking about the decoupling of related papers "their decoupling seem shallow" is itself a shallow explanation. I think it would be better to explain in technical terms the difference between their decoupling and the decoupling of this paper.
    - Minor
        - I would potentially edit the title to "complex systems" or "a complex system"
        - end of page 1 (e.g., ) empty two times
        - just after equation 3 it says dx(t) = ... yet dx(t) doesn't seem to appear in the equation.

**Strength And Weaknesses:**

- Strengths
    - The problem of modeling modular dynamical systems at irregular time intervals is compelling.
    - The paper does a good job introducing neural ODEs, neural controlled dynamics and self-attention.
    - The paper evaluates in a wide diversity of scenarios, capturing different applications of the proposed approach.
- Weaknesses
    - Experimental results are not strong and some better results are missing from the table of comparisons. In particular, at least two methods achieve much better results in the charged and springs datasets. The paper obtains 68% link prediction accuracy in charged and 95.4% on springs. However, AFAIK, this is on the same datasets that Neural Relational Inference with Fast Modular Meta-learning (NeurIPS '19) obtained 88.4% and 99.9%, respectively. Furthermore, the paper where these datasets come from (Neural Relational Inference for Interacting Subsystems, ICML '18) is not shown in the table of comparisons, and that also obtains significantly better results: 82.1% and 99.9%.
    - The proposed architecture is missing justification, in the sense that it is proposed as something we can do, but the reason why the added complexity is necessary is missing. Better ablations in the experimental section, improved explanations in the method section and a better interplay between the two sections would greatly help in this direction.
    - Although good for many other conferences, the level of novelty (mainly combining attention-based mechanisms with neural controlled dynamics), is IMO not enough for the ICLR bar.

**Summary Of The Paper:**

This paper aims at modeling complex dynamical systems into smaller subsystems. It does so by combining neural controlled dynamics(Kidger et al.) with self-attention (Vaswani et al.). Empirical evaluations are performed in multiple regression and link prediction environments from 3-body gravitation, springs, charged particles and human actions.

**Summary Of The Review:**

I like the goal of this paper and believe it has many practical applications. At the same time, I feel that, at the moment, the methodology and experimental results do not support the novelty and quality of the proposed approach.

---

> ### Author Response · Authors · 2022-11-17
> **response to reviewer 2**
>
> We thank reviewer 2 for his/her constructive advice. Responses to some other concerns are listed below.
>
> We believe that there is some misunderstanding of the basic setting of the paper, which is potentially due to our writing. We explained our challenging setting with an example in overall response 1.9. We further emphasize this in the revised manuscript. Thank you for pointing this out.
>
> Besides, we also presented more experimental results supporting our claim in the overall response as well as the revised paper.
>
> Thank you again and looking forward to your further comments.

---

### Official Review · Reviewer_ateH · 2022-11-01

**Confidence:** 5
**Correctness:** 3
**Technical Novelty And Significance:** 3
**Empirical Novelty And Significance:** 2
**Recommendation:** 3

**Clarity, Quality, Novelty And Reproducibility:**

The originality is good, but the experiments do not support it well. The presentation is overall good.

**Strength And Weaknesses:**

Strong points.
1. The proposed method and the basic idea make sense to me. The design of the meta-system is novel.
2. The method is described well, and the paper presentation is nice.
3. The results show the best performance of the proposed DNS.

Weak points.
1. The experimental settings require improvement. There is no sufficient hyper-parameter study of the baselines. Thus, the comparisons among the models may not be fair, which leads to unreliable conclusions.
2. There is no study about the complexity. Compared with other baselines, which also work in a data-driven manner, the complexity of the proposed method is not clear. It seems the parameter size is far larger compared with baselines, and thus the computation time is a concern.
3. The performance improvement of the proposed method seems to be not steady. From the results, we can observe that the performance is very sensitive to the parameter size (influenced by the hyperparameter). Therefore, it is also concerned about the generalization ability of the proposed method.

**Summary Of The Paper:**

This paper solves the problem of decoupling dynamical complex systems in a data-driven manner. The authors propose an improved version of the projected differential equations. The authors conduct experiments on various important problems with synthetic or real-world datasets. The results show that the proposed DNS can achieve the best performance.

**Summary Of The Review:**

The experimental part requires a lot of improvements, although the idea makes sense.

---

> ### Author Response · Authors · 2022-11-17
> **response to reviewer 1**
>
> We thank reviewer 1 for his/her constructive advice. Responses to some other concerns are listed below.
>
> For the experimental part, we have conducted more experiments with rich hyperparameter settings as suggested. Please refer to the overall response for more details.
>
> Thank you again for identifying the novelty of our work, and we are looking forward to your further comments.

---

### Author Response · Authors · 2022-11-17
**response 1**

In the sequel, we refer the reviewers ateH, koo9, gG1Z, gYdf as reviewer 1, 2, 3, 4, respectively.

We thank all the reviewers for their contributing suggestions and great effort in providing reviews timely. We also thank reviewers 1, 3, and 4 for recognizing the novelty of our work. In general, we want to emphasize that the novelty is not only from the proposed approach, it also arises from the challenging and meaningful problem: learning to decouple a complex system without physical entity correspondence. To the best of our knowledge, there have been no previous attempts.

In this post, we give general feedback on the common concerns raised by the reviewers. Some minor responses to each of the reviewers will be posted following the comment sections.

We also updated the revised version of our submission correspondingly.

1 Experiment:

We note that most of the reviewers’ concern is on the experimental part, we give our feedback as follows. We also conducted more experiments suggested by the reviewers to support our claim.

1.1 Hyperparameter and fairness

In ALL the experiments, we performed the grid search on hyperparameters for ALL the selected counterparts as well as our method (see Table 1, 2, 3, 4). All these hyperparameter settings are under the consideration of fairness, such that each setting of our method corresponds to another setting of the selected counterparts with comparable model capacity. This is also the reason why we reported model size (highly related to model capacity) in Table 3. To be more convincing, we also added two additional settings (CT-GRU 1024 and NeuralCDE 1024) in Table 3 in the revised manuscript, while their performance is far below DNS. We believe such a comparison is fair, self-consistent and sufficient for supporting our claim.

1.2 Complexity and efficiency

In general, the complexity of the proposed method is very similar to that of NeuralCDE. Each subsystem shares the same function F in Eq. (11), but only differs on the control x_i. While the main computational burden in Eq. (11) (as well as the whole DNS) is at F, x_i is obtained using a simple 2-layer MLP. Additionally, the evolution of interaction A is also light-weight, consisting of (matrix) parameters Q and K of which one dimension is equal to the number of subsystems (typically <= 10). In summary, compared to a standard NeuralCDE, our approach is not with significant higher model complexity. We therefore believe that is not demanding to concern about the extra computational burden.

1.3 Sensitivity

We appreciate reviewer 1 for pointing out the sensitivity issue in Table 3. However, we emphasize that such sensitivity issue occurs ONLY at noisy case under limited settings when controls are injected with noise. Actually, we also discussed this sensitivity (in the initial submission) and proposed a variant DNS_S to stabilize the performance. According to Table 3, DNS_S can achieve this challenging goal.

1.4 Reproducibility

We have provided the code (in supplementary material) for reproducing ALL the experiments reported in our initial submission. We strongly encourage the reviewers to test that, as suggested by reviewer 4.

1.5 Gradient stability

In fact, we did not discuss the gradient explosion/vanishing problem in the initial submission. Instead, our claim is “more stable gradient in the early training stage”to imply that such a formulation of A allows for faster convergence and potentially better performance. According to reviewer 3, this statement indeed may cause misunderstanding. Therefore, we re-organize the language in the revised version into “gradient with higher magnitude in the early training stage, which allows for faster convergence and more significant gradient backpropagation”.  Detailed ablation study can be found in response 1.7.

---

> ### Author Response · Authors · 2022-11-17
> **response 2**
>
> 1.6 Cross-validation
>
> As suggested by the reviewers, we conducted additional 5-fold cross-validation on spring dataset, and the results are summarized as follows with mean accuracy and std. We can simply conclude that DNS achieved prominent performance among all the methods. Specifically, we note that the performance of CT-RNN 256 under irregular setting dropped drastically compared to the result without 5-fold cross-validation (81% in Table 3). It seems that CT-RNN 256 is quite sensitive to the selection of training set.
>
> |            |        regular	|           irregular|
> |---|---|---|
> |CT-RNN	|74.58+2.42	|53.26+1.01|
> |CT-GRU	|90.36+0.18	|89.79+0.13|
> |NeuralCDE|	91.08+0.12|	90.52+0.11|
> |RIM	|87.45+0.08	|-|
> |DNS	|94.76+0.58	|93.70+2.14|
>
> 1.7 Ablation
>
> Since our method merely incorporates an extra meta-system and a control encoder for modeling the interaction compared to standard NeuralCDE, we conduct experiments under different settings to see how different encoders and hidden state dimension can contribute to improve NeuralCDE. Results are summarized as follows. We see that with an extra control encoder and varying hidden dimensions, there is no obvious performance difference among these settings. However, once the interaction meta-system is imposed, DNS can achieve quite significant performance gain. This in turn shows the necessity of the proposed meta-system for explicitly modeling the evolving interactions.
>
> |control settings (encoder + system dim)	|Accuracy(%)|
> |---|---|
> |No encoding + 128	|90.02|
> |No encoding + 256	|91.06|
> |No encoding + 512	|91.57|
> |MLP(dim=2*input size) + 128	|87.01|
> |MLP(dim=2*input size) + 256	|90.87|
> |MLP(dim=2*input size) + 512	|91.51|
> |MLP(dim=16*input size) + 128	|91.17|
> |MLP(dim=16*input size) + 256	|91.08|
> |MLP(dim=16*input size) + 512	|90.70|
> |DNS 8 512	|95.38|
>
> 1.8 Missing entries in Tables
>
> We figured out to train most of the failed settings (in the initial submission) for the selected counterparts and completed the unfinished experiments. All the results are updated in the revised manuscript. The new results consistently support our claim about the advantages of the proposed method.
>
> 1.9 Performance compared to Neural Relational Inference (NRI)
>
> We emphasize that the setting in our experiments is totally different from that of NRI. In our setting, the correspondence between features and physical entities is unknown. For example, in three body test the control is 9-dimensional, but the model does not know the grouping of the features (e.g., 3-3-3 dim corresponding to 3 entities with 3-dim each). Instead, this 9-dim feature is treated as a whole for further decoupling. However, in the setting of NRI, such grouping is known by the model. This is also the key reason why this problem is so challenging, as pointed out by reviewers 3 and 4.
>
> 2. Novelty and contribution
>
> Once again, learning to decouple a complex system without physical prior is a challenging task, and has barely been investigated in previous works. As commented by reviewers 1, 3, and 4, the task itself is novel and challenging. Our approach is not a simple combination of NeuralCDE and attention mechanism. As neither NeuralCDE nor attention can handle this task, our approach contributes to proposing a novel solution to achieve this from a system decoupling perspective. NeuralCDE and attention merely serve as two tools in our modeling. We believe this could provide some insights into how a complex system can be viewed and modeled for other researchers. Besides, we made the first attempt to interpret evolving interactions as projected differential equations under a machine learning context. We will clarify this in the revised version to avoid further misunderstanding.

---

### Decision · Program_Chairs · 2023-01-20

**Decision:**

Reject

**Justification For Why Not Higher Score:**

The decision is based on agreeing with the majority of the reviewers.  The experimental section was improved, but not at a level which merits acceptance, when compared to other papers.  Then, the paper is just incremental.

**Justification For Why Not Lower Score:**

n/a

**Metareview: Summary, Strengths And Weaknesses:**

The paper focuses on the problem of modeling complex dynamical systems by multiple, independent ones.  The approach is incremental, and is supported by an eclectic experimental section.

Despite the good rebuttal phase, the experimental section still can be improved, for instance by adding confidence intervals to get better insight in the significance.  Also, in some cases the method is not superior or seems not to be tested with sufficient fairness (e.g., increased size of CT-GRU may improve).

All in all, the paper needs strengthening to be beyond incremental.